# Comparison of Different Methods for Extracting the Astaxanthin from *Haematococcus pluvialis*: Chemical Composition and Biological Activity

**DOI:** 10.3390/molecules26123569

**Published:** 2021-06-11

**Authors:** Yicheng Tan, Zhang Ye, Mansheng Wang, Muhammad Faisal Manzoor, Rana Muhammad Aadil, Xinghe Tan, Zhiwei Liu

**Affiliations:** 1College of Food Science and Technology, Hunan Agricultural University, Changsha 410128, China; yichengtan@hunau.edu.cn (Y.T.); ye274919161@163.com (Z.Y.); 2Institute of Bast Fiber Crops, Chinese Academy of Agricultural Sciences, Changsha 410205, China; wangmansheng@caas.cn; 3School of Food and Biological Engineering, Jiangsu University, Zhenjiang 212013, China; faisaluos26@gmail.com; 4National Institute of Food Science and Technology, University of Agriculture, Faisalabad 38000, Pakistan

**Keywords:** *Heamatococcus pluvialis*, astaxanthin, cell disruption techniques, antioxidant activities, anti-proliferative ability

## Abstract

In this study, the impact of different cell disruption techniques (high-pressure micro fluidization (HPMF), ionic liquids (ILs), multi-enzyme (ME), and hydrochloric acid (HCl)) on the chemical composition and biological activity of astaxanthin (AST) obtained from *Haematococcus pluvialis* was investigated. Results indicated that all cell disruption techniques had a significant effect on AST composition, which were confirmed by TLC and UPC^2^ analysis. AST recovery from HCl (HCl-AST) and ILs (ILs-AST) cell disruption techniques was dominant by free and monoesters AST, while AST recovery from HPMF (HPMF-AST) and ME (ME-AST) cell disruption techniques was composed of monoesters, diesters, and free AST. Further biological activity analysis displayed that HCl-AST showed the highest ABTS and DPPH activity, while ILs-AST showed better results against the ORAC assay. Additionally, ILs-AST exhibits a stronger anti-proliferation of HepG2 cells in a dose-dependent manner, which was ascribed to AST-induced ROS in to inhibit the proliferative of cancer cells.

## 1. Introduction

Microalgae is a natural and renewable source of protein, lipids, pigment and other valuable components, which has attracted significant research interest around the world for algae cultivation for biomass accumulation and extraction [1,2]. Astaxanthin (AST) as a symmetric ketocarotenoid is widely distributed in microalgae and shells of crustaceans [3]. It has been extensively applied as a nutritional supplement in foods, feeds, and pharmaceuticals [4]. *Haematococcus pluvialis* as the most abundance and valuable source of natural AST, which is acknowledged as a “super antioxidant” [5]. AST exists in *H. pluvialis* in three forms diesters (~25%), monoesters (~70%), and free (~5%) [6]. However, the elaborated three-layer cell wall of H. pluvialis enhances the chemical and mechanical resistance, hindering the AST recovery.

A significant effort has been devoted to break the tough cell wall of *H. pluvialis* using some strategies, e.g., the conventional method (bread milling and high-pressure homogenization), innovative techniques (ultrasound (US), pulsed electric field (PEF), multi-enzyme (ME), hydrochloric acid (HCl), high-pressure micro-fluidization (HPMF), and ionic liquids (ILs) [1,2,7,8,9]. Liu et al. [1] reported that more than 80% AST was recovered from *H. pluvialis* after being pretreated by ionic liquid (1-butyl-3-methylimidazolium chloride) under mild conditions (pretreatment with 40% IL aqueous solution at 35 °C, followed by methanolic extraction at 50 °C). Zhang et al. [8] have achieved 80.52 ± 2.28% and 71.08 ± 2.49% AST recovery from *H. pluvialis* after pretreating with HCL multiple enzymes, respectively. The fundamental acknowledgement of breakdown of the cell wall of *H. pluvialis* for AST recovery has been well established. However, study regarding the influence of the extraction method and cell disruption techniques on the structure and composition of AST obtained from *H. pluvialis* and its biological activity is rare [8].

For biological activity, AST can significantly decrease oxidative stress and free radicals. AST has several nutraceutical and human health-promoting applications including in the anti-proliferation, anti-inflammatory, anti-cancer, and cardiovascular systems [5,10,11,12]. With increasing demand, AST will be one of the important microalgal products in the future [5]. AST is known as a potent antioxidant that may play an inhibitory role in all stages of the cancer cell. Mularczyk, Michalak [13] reported that AST exhibited notable antitumor activity compared to the other carotenoids like β-carotene and canthaxanthin. Palozza, Torelli [14] described that *H. pluvialis* extract repressed the colon cancer cell’s growth by preventing cell cycle progress and increasing apoptosis. Some chemotherapy agents have been suggested to induce the production of ROS, through further increasing the ROS in cancer cells, killing the cancer cells, which has become one of the strategies for cancer treatment [15,16].

Based on our previous studies which have elucidated that the efficiency and mechanism of different cell disruption techniques (HPMF, ILs, ME, and HCl) on the recovery of AST from *H. pluvialis* [1,2,3,17,18], this study aims to further explore the impact of those cell disruption techniques on the composition and biological activity of the AST obtained. The composition of AST was characteristic of thin-layer chromatography (TLC) and ultra-performance convergence chromatography (UPC^2^). Additionally, the anti-oxidation and anti-proliferation activities of AST were also investigated.

## 2. Results and Discussion

### 2.1. TLC and UPC^2^ Analysis

Figure 1 describes that the effect of different cell disruption techniques on the composition of AST was analyzed by TLC. The robust structure of the trilaminar cell wall of *H. pluvialis* makes this microalga remarkably resistant to chemical and physical cell disruptions and complicates the extraction process of AST from these cyst cells. The IL, ME and HCL chemical cell wall breakdown techniques show a great potential for cell wall disruption and high-efficiency AST extraction. Results indicated that the profile of AST was significantly affected by all cell disruption techniques. It is well known that there are three kinds of AST which exist in *H. pluvialis* including free AST, monoester AST and diester AST [13]. Among different cell break techniques, HCl-AST contains the most free AST, while the HPMF-AST, ILs-AST, and ME-AST have abundant amounts of the ester. Sachindra et al. [19] stated that AST and its diesters and monoesters (63.5–92.2%) were the chief carotenoids in two important deep-sea species *Solonocera indica* and *Aristeus alcocki*. The R_f_ values noted for AST diester and monoester were in line with the outcomes reported by Kobayashi and Sakamoto [20]. Sila et al. [21] also reported the similar Rf values for the three bands such as AST (0.33), AST monoester (0.66), and AST diester (0.80) extracted from pink shrimp shell waste.

Further analysis conducted by UPC^2^ was presented in Figure 2. UPC^2^ analysis exhibited that the major peaks for HCl-AST appeared at 2.557 and 2.813 min (Figure 2b). The profile of HCl-AST contains free AST with few AST esters. Our results demonstrated that HCl pretreatment can effectively hydrolysis the ester bond and most of the AST exists in non-esterified form. Additionally, observed in Figure 2c–e, that AST derivatives were the main components, but the chemical composition was not the same in HPMF-, ILs-, and ME-AST. Figure 2c–e showed that the carotenoid pattern was modified and suggesting a possible difference in the solubility of individual components under the different processing methods. Figure 2c indicated that the peaks for ILs-AST appeared at 2.273, 2.528, 2.814, 3.733, 6.270, 6.853, 6.966, and 8.301 min. Our findings are in line with Miao, Lu [22] who investigated the AST esters in *H. pluvialis* but, in our study, the components of four AST extracts were significantly different from each other due to different processing conditions. The relative proportion, composition, and several AST molecular species, in the available literature and this work, are different possibly due to different extraction methods.

### 2.2. DPPH and ABTS Analysis

Antioxidant activity was investigated in a DPPH radical scavenging assay, which is regarded as the first approach for evaluating antioxidant activity due its simplicity, speed, and low cost. In principle, the hydrogen-donating ability of antioxidants reduces the free radical DPPH (purple) to a stable DPPH (yellow), leading to decreased absorbance at 517 nm [23]. For the DPPH assay, the radical scavenging activities of HCl-AST (IC_50_ = 15.39 µg/mL) were significantly higher than those of ILs-AST (IC_50_ = 43.81 µg/mL), HPMF-AST (IC_50_ = 52.76 µg/mL) and ME-AST (IC_50_ = 56.25 µg/mL) (Figure 3a). Earlier, Dong, Huang [24] reported that the AST extracted from *H. pluvialis* by HCl acid pretreatment followed by acetone extraction exhibited a stronger reducing power and DPPH free radical-scavenging ability than hexane/isopropanol, methanol followed by acetone extraction, and soy-oil extraction methods. Moreover, Sila, Ayed-Ajmi [21] reported that the AST extracted from pink shrimp shell waste was exhibited as the strongest radical scavenger with an IC50 of 15.87 µg/mL. Furthermore, Sindhu and Sherief [25] stated that the DPPH activity of AST was much better than carotenoids obtained from red shrimp (*Aristeus alcocki*). Sidhu and Sherief [25] described that the higher proportion of AST diester and poly unsaturated fatty acids (PUFAs) in the carotenoid extract obtained from *Aristeus alcocki* shell were responsible for higher antioxidant activity. Thus, the powerful antioxidant property of carotenoid extract may be attributed to the antioxidant synergism of AST and PUFAs present in the extract.

The antioxidant activity of extracted AST through different cell disruption methods were assessed by ABTS^+^ analysis as presented in Figure 3b. ABTS radical scavenging activities of HCl-AST (IC_50_ = 20.32 µg/mL) were slightly higher as compared to those of ILs-AST (IC_50_ = 21.73 µg/mL), HPMF-AST (IC_50_ = 22.09 µg/mL) and ME-AST (IC_50_ = 25.53 µg/mL). According to TLC and UPC^2^ results, AST dominant by free AST and had a higher DPPH and ABTS radical scavenging activities as compared to those of monoester AST and diester AST. Earlier, Jaime, Rodríguez-Meizoso [26] stated that the hydroxyl groups after being esterified by fatty acid will change the antioxidant activity of AST. They further described that the fatty acid has been hydrolyzed and the TEAC value was increased (0.168 to 0.434 mmol Trolox/g). The antioxidant activity of the AST extracts from *H. pluvialis* after hydrolysis was higher than the original extract. It indicates a higher contribution of unmodified hydroxyl groups to the delocalization of electrons in the AST molecule. Liu and Osawa [27] stated that AST comprises a unique molecular structure in the presence of keto and hydroxyl moieties on each ionone ring, which is accountable for the strong antioxidant attributes.

### 2.3. ORAC Analysis and PSC Analysis

The ORAC assay is an antioxidant assay based on a fluorescent indicator that depends on both the inhibition degree and inhibition time. Figure 3c shows ORAC kinetics curves of AST extract by different methods. As indicated in the ORAC assay the oxygen radical scavenging capacities of four AST samples indicated that all the four AST forms can quench with onsite produced ROO^·^ and delay the damage of the fluorescent probe by peroxyl radicals. It is noticeable that the capability to impede free radical damage by 1 μg/mL ILs-AST was higher than that in the other three AST. The relative Trolox equivalent ORAC value is shown in Figure 3c. The relative ORAC value of ILs-AST was increased from 35.6 ± 0.81 μM Trolox to 110.4 ± 1.02 μM Trolox when the concentrations of ILs-AST increased from 1 to 4 μg/mL. According to our TLC and UPC^2^ results of ILs-AST, ILs-AST contains a large amount of AST-monoester. The reason fir which AST-monoester shows a stronger antioxidant than AST-diester and free AST may be due to the high electron donation activity of one hydroxyl group esterified with fatty acid to the ROO^·^, thus terminating the peroxide chain reaction. Earlier, Régnier, Bastias [28] described that the ORAC value for chemical extraction and high-pressure processing was higher than the Dimethyl Sulphoxide (ORAC value: 8.1 µM TE/g of AST) on extractions from *H pluvialis,* because the natural extracts comprising esters exhibited more powerful antioxidant activities than free AST.

The PSC kinetics curves of were AST extracted with different methods as shown in Figure 3d. These results specified that all the AST samples could scavenge ROO·, produced from thermal degradation of AAPH (2, 2′-Azobis (2-methylpropionamidine) dihydrochloride) that oxidizes DCFH to fluorescent dichlorofluorescein. Trolox (3.125–50 μM) inhibited the oxidation of DCFH by ROO· and was utilized as a standard to formulate a calibration curve (y = 0.0026x + 0.2297, R^2^ = 0.9962). AST-ME showed a relatively high free radical-scavenging activity against ROO·, and the PSC values were equivalent to 30.37 ± 1.49 Trolox (Table 1). The results differ from ORAC, which may be due to sensitivity and types of fluorescent probes, in which the active oxygen scavengers and peroxyl radical scavengers of AST-E were significantly higher than F-AST.

### 2.4. Anti-Proliferative Activity and Intracellular ROS

Recently, increasing interest in some natural compounds and novel pharmaceuticals has attracted the consideration of researchers. In particular, marine compounds with notable antineoplastic characteristics are analyzed for their numerous apoptotic impacts on different tumor cell types while normal cells seem to exhibit low toxicity [29]. AST has exhibited anticancer activity through various mechanisms such as apoptosis induction, cell growth inhibition, and interference of cell cycle progression [30]. Zhang and Wang [30] reported the strong anti-oxidative effects of AST against multiple kinds of cancer, through a variety of mechanisms. In this sense, the chance to prove the same power on HPMF-, ILs-, ME-, and HCl-extracted AST is of great concern for the possibility to address the utilization of seafood in nutraceutical applications. In this study, we have inspected the anti-proliferative effect of AST extracted from *H. pluvialis* through these four techniques in a HepG2 cell line. Dose-dependent viability was determined to increase the concentration of AST. As shown in Figure 4, the highest anti-proliferative activity was observed in ILs-AST treatment after 24 h, which contains a large quantity of AST monoesters. As the time extended to 48 h, the proliferation ability of HepG2 was further reduced. AST can persistently reduce the proliferation of HepG2 cells, in which AST monoesters show better anti-proliferative activity. In HepG2 cells, an exceptional arrest induction at the G0/G1 phase was observed after 24 h incubation of AST treatment at 25 μM and 42 μM, exhibiting a potent decrease in cells in the S and G2-M phases [31]. Commonly, the cell cycle progression is regulated by a set of cyclin-dependent kinase (CDK). They are threonine/serine kinases that regulate cell cycle progression and change their catalytic actions by interplay with CDK and cyclins inhibitors. Variations in the expression of CDKs in malignant cells, under-expression of CDK, and overexpression of cyclins inhibitors have been commonly described. The renovated CDK activity describes a selective benefit for cancer cell growth. Numerous investigations described that AST influences the activity of CDK, inhibiting tumor progression. AST as carotenoid compounds has been stated to inhibit cyclin D1 and tumor cell growth-stimulating p21 [32]. As cyclin D1 controls G1 to S-phase shift, its suppression may be due to the delay in tumor cell line proliferation. A CDK inhibitor such as p21 is responsible for G1 arrest and, consequently, cell cycle arrest [33].

In the CCK-8 assay, different AST concentrated samples inhibited the process of cell proliferation. When the concentration of AST reached 16 μg/mL, the cell proliferation was significantly decreased. So, AST was shown to be toxic, as 16 μg/mL of AST treatment led to a reduction in the proliferation of HepG2 cells. ILs-AST containing more AST monomers show better anti-proliferative activity. These findings were supported by the study of Messina, Manuguerra [34] who observed the anti-proliferative activity of carotenoid AST extracted by supercritical fluid extraction from shrimp by-products. It was observed that pre-treating normal fibroblast cells with AST increased the cell viability in a dose-dependent manner attesting its antioxidant power. Furthermore, Nagaraj, Rajaram [31] analyzed the cell viability by MTT assay with different concentrations of AST (5, 10, 15, 20, and 25 µg/mL) to incubate at different periods. The results indicated that the AST (25 µg/mL) significantly obstructs the HepG2 cancer cell growth and induces cell apoptosis.

Recent studies have elucidated that homeostasis between autophagy and mitochondrial function is vital to the physiological activity of cancer cells, which is helpful for the therapy performed tumors by the drugs. It is well known that mitochondria not only provide energy, but also regulate cell death [35]. Recent papers reported that ROS which is mainly produced by mitochondria has a crucial factor to promote AMPK signaling for autophagy [36,37]. To better understand the inhibitory mechanism of each sample in/of HepG2 cells, the accumulation of ROS in HepG2 after four AST treatments were investigated. As shown in Figure 5 and Figure 6, the accumulation level of ROS was significantly increased after being incubated in different AST samples (*p* < 0.05). Among these, ILs-AST was more effective for increasing the ROS levels, while the ROS accumulation level was further increased after the increase in AST concentration. These results were consistent with our anti-proliferative activities. Several studies have reported the pro-oxidant effects of some carotenoids on cancer cells with the production of ROS. Kim, Heo et al. [38] have reported that the growth inhibition in leukemia cell lines by fucoxanthin, which was attributed to ROS generation by fucoxanthin, leading to leukemia cell apoptosis. Additionally, Zhang, Zhou [39] have found that *dendrobium offcinale* polysaccharides significantly inhibited the proliferation of colon cancer cell line CT26 and elevated autophagy level. Further studies unveiled that *dendrobium offcinale* polysaccharides disrupted mitochondrial function through increasing reactive oxygen species (ROS) and reducing mitochondrial membrane potential (MMP), resulted in impaired ATP biosynthesis, which activated AMPK/mTOR autophagy signaling. Therefore, AST may also exhibit its anti-cancer effects through activation of ROS. Further studies are needed to clarify its mechanism.

## 3. Materials and Methods

### 3.1. AST Preparation

AST (Bayou Biogenic Co. Ltd. Yunan, China) was obtained from different cell disruption techniques (HPMF, ILs, ME, and HCl) as prepared as following our previous research [1,2,8].

### 3.2. TLC Analysis

AST was analyzed from TLC aluminum sheets with 10 × 20 cm precoated with the silica gel 60 (Merck Ltd., New Delhi, India). AST was dropped on a TLC sheet and run in acetone:hexane (3:7) solvent [31].

### 3.3. Ultra-Performance Convergence Chromatography (UPC^2^) Analysis

AST was performed in the Acquity UPC^2^ System (Waters Corp, Milford, MA, USA) suited with a PDA detector. Then, the X-Bridge BEH-2-EP with a 1.7 μm 3.0 × 100 mm column was utilized. The temperature of the column was set at 35 °C. A mobile phase comprising CO_2_ (A) and CH_3_OH (B) was used, and the flow rate of this column was set at 1.0 mL/min. AST separation was attained through a gradient between solvents for 10 min as follows: B was run for 3 min at 10% and continued for 3 min, and 10–20% for 3 min, and kept for 5 min. The isolated AST esters were analyzed through photodiode arrays (PAD). The absorbance was measured at 457 nm, the reference of compensation from 530 to 600 nm.

### 3.4. Determination of Antioxidant Properties of H. Pluvialis

#### 3.4.1. DPPH Scavenging Activity

The DPPH radical scavenging activity of AST was determined by using the method described by Ren, Chen [40] with slight modification. Firstly, 3 mL of 0.1 mM DPPH in methanol were mixed with 0.5 mL of samples at different concentrations (4, 8, 16, 32, 64 and 128 µg/mL). The blank consisted of 3 mL DPPH and 0.5 mL methanol. At the end, the absorbance of samples was measured at 517 nm by a spectrophotometer (ASONE V1100D, Osaka, Japan) after 30 min. The rate of DPPH radical scavenging activity was calculated by Equation (1), and IC_50_ was indicated as the concentration of 50 % of DPPH radical scavenging activity.
$$\mathrm{DPPH}\;\mathrm{scavenging}\;\mathrm{rate}\;(\%)=(1-\frac{A_{1}-A_{2}}{A_{0}})\times 100$$
where A_0_ was the absorbance of the DPPH solution without sample, A_1_ was the absorbance of the sample mixed with DPPH solution, A_2_ was the absorbance of the sample without DPPH solution.

#### 3.4.2. ABTS Scavenging Activity

The ABTS radical scavenging activities of AST was using the method described by Xu, Xu [41] with slight modification. ABTS solution (7 mM) was prepared by dissolving 0.192 g 2, 2′-azino-bis (3-ethyl-benzothiazolie-6-sulfonic acid), 0.33 g potassium persulfate in 0.1 L of distilled water. 0.3 mL sample solution with different concentrations (4, 8, 16, 32, 64 and 128 µg/mL) was added into 2.7 mL ABTS solution. The blank consisted of 2.7 mL ABTS solution and 0.3 mL methanol. The absorbance of samples at 734 nm was measured by a spectrophotometer after 10 min. The rate of ABTS scavenging activity was calculated by Equation (2), and IC_50_ was indicated as the concentration of 50 % of ABTS radical scavenging activity.
$$\mathrm{ABTS}\;\mathrm{scavenging}\;\mathrm{activity}\;(\%)=(1-\frac{A_{1}}{A_{0}})\times 100\%$$
where A_0_ was the absorbance of ABTS solution without sample, A_1_ was the absorbance of the sample mixed with solution.

#### 3.4.3. Oxygen Radical Absorbance Capacity (ORAC)

ORAC assay was completed using the method described by Liu, Luo [42] with slight modification. Briefly, 150 µL of fluorescein (50 mM) was mixed with 25 µL of sample and incubated at 37 °C for 30 min. After that, 25 µL of AAPH (50 mM) was added to each well and fluorescence measurements were taken over a 5 h time. To build the blank decay curve and Trolox standard decay curve, 25 µL of phosphate buffer or Trolox standard solution was added instead of the sample solution. The results of the sample were calculated by using a regression equation between the Trolox concentration (3.125, 6.25, 12.5, 25, 50, 100 µm M) and the net area under the fluorescein decay curve. The ORAC values were expressed as µM of Trolox equivalent (TE) per gram of dried sample. The area under the fluorescence decay curve (AUC) was calculated by Equation (3):AUC = 1 + f_1_/f_0_ + f_2_/f_0_ + f_3_/f_0_ + …… + f_299_/f_0_ + f_300_/f_0_(3)
where F_0_ is the initial fluorescence at 0 min and F_i_ is the fluorescence reading at I min.

#### 3.4.4. Peroxyl Radical Scavenging Capacity (PSC)

The PSC assay was using the method described by Adom and Liu [43] with slight modification. Briefly, 80 µL of DCFH-DA (2.48 mM) was hydrolyzed with 900 µL of KOH (1.0 mM) for 3–5 min to remove the diacetate (DA) moiety and then diluted to 6 mL total volume with phosphate buffer (75 mM, pH 7.4). An amount of 150 µL of fluorescein (50 mM) was mixed with 25 µL of sample and incubated at 37 °C for 30min. An amount of 25 µL samples were mixed with 150 µL of DCFH and incubated at 37 °C for 30 s. After that, 25 µL of AAPH (50 mM) was added to each well and fluorescence measurements were taken over a 40 min time. The fluorescence was set at 485 nm as the excitation wavelength and 538 nm as the emission wavelength. To build the blank decay curve and Trolox standard decay curve, 25 µL of phosphate buffer solution or Trolox standard solution was added instead of the sample solution. The areas under the average fluorescence–reaction time kinetic curve (AUC) for both controls and samples were integrated and used as the basis for calculating antioxidant activity, which was calculated by Equation (4)
PSC unit = 1 − (SA/CA)(4)
where SA is AUC for sample or standard dilution and CA is AUC for the control reaction using only buffer. The median effective concentration (IC50) was defined as the dose required to cause a 50 % inhibition (PSC unit = 0.5). The PSC values were expressed as µM of ascorbic acid equivalents (AEE) per 100 g of dried sample.

### 3.5. Anti-proliferative Activity

HepG2 cells (passage-16–25, Vishay Biological Co., Ltd., Changsha, Hunan, China) were continued in DMEM with 10 % FBS, 1 % penicillin–streptomycin. CCK-8 assay was utilized to evaluate the Anti-proliferative activity of AST on the HepG2 cells. Cytotoxicity measurement was done before the CCK-8 assay. When the cells enter into the logarithmic growth phase with the confluence of 80%, 10 µL AST samples (2, 4, 8, 16, and 32 µg/mL) were added into wells and incubated with 5% CO_2_ at 37 °C for 24 h and 48 h, respectively. Then, 10 µL per well of CCK-8 was inserted in the cells incubated for another 4 h. Finally, the absorbance was monitored at 450 nm using a Microplate reader (Bio-Tek ELX50, Winooski, VT, USA). Cell proliferation inhibition was counted according to the manufacturer’s procedure.

### 3.6. ROS Measurement

Accumulation of ROS was performed using the method described by Dai, Li [44]. Briefly, IL-AST, ME-AST, and HPMF-AST (2, 4, 8, 16, and 32 µg/mL) were incubated with HepG2 cells. The cell cultured in DMEM acted as a negative control. After 24 h incubation, cells were cultured with DCFH-DA for 30 min. After obtaining supernatants, washing of cells one time with PBS was carried out. Finally, fluorescence was detected by flow cytometry. The outcomes are shown as mean ± SD of comparative fluorescence units of three autonomous tests in total.

### 3.7. Statistical Analysis

All the tests were performed thrice, and outcomes are shown as mean ± standard deviation. All of the analyses were accomplished through OriginLab 8 software (Northampton, MA, USA). The outcomes were estimated from Duncan’s posthoc methodology. The adopted significance level was *p* < 0.05.

## 4. Conclusions

This study employed different cell disruption techniques (HPMF, ILs, ME, and HCl) for the first time for the extraction of AST from *H. pluvialis*. Results described four samples containing different ratios of monoester, diesters, and free-AST. Importantly, HCl-pretreated AST was dominated by free AST which exhibited stronger antioxidant activities (DPPH and ABTS scavenging activity). ILs-pretreated AST containing the maximum level of monoester AST exhibits a stronger total antioxidant capacity and anti-proliferation. The reason for the inhabitation of proliferative cancer cells may be ascribed to the production of ROS induced by AST. These techniques have a significant impact on chemical composition and biological activities. Still, variations in the mode mechanism in the chemical composition of extracted AST by these techniques should be examined for further knowledge in this field. Our verdicts propose that AST obtained from *H. Pluvialis* via HPMF, ILs, ME, and HCl holds significance in nutraceutical and pharmaceutical industries and affords an efficient and environmentally friendly way to obtain AST.

## Figures and Tables

**Figure 1 molecules-26-03569-f001:**
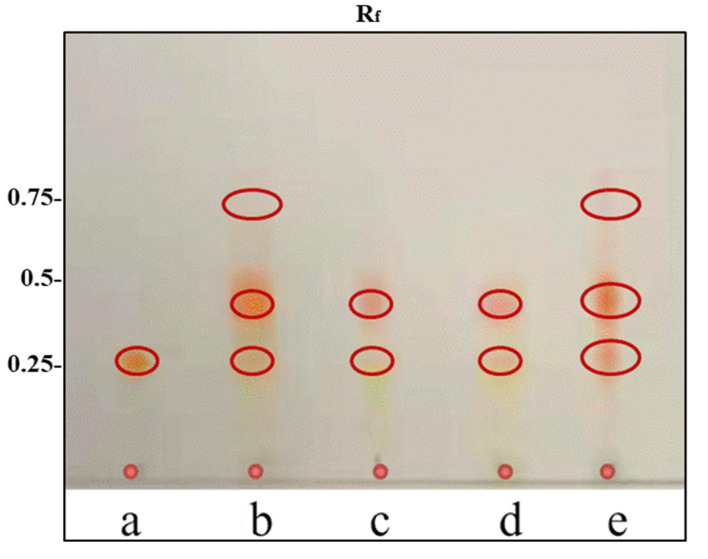
TLC analysis of AST extracted from *H. pluvialis* by different cell disruption techniques. (**a**) AST standard. (**b**) HPMF-AST. (**c**) ILs-AST. (**d**) ME-AST. (**e**) HCl-AST.

**Figure 2 molecules-26-03569-f002:**
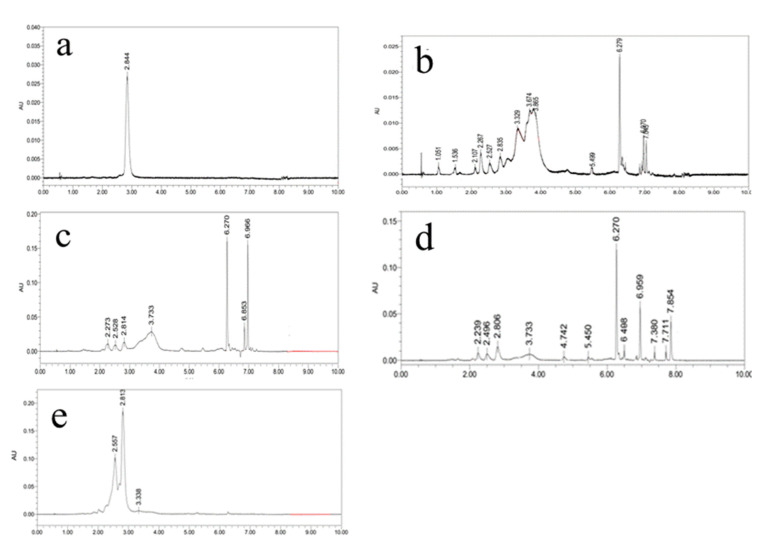
Analytical UPC^2^ chromatograms of red phase *H. pluvialis* cells’ extracts. (**a**) AST standard. (**b**) HPMF-AST. (**c**) ILs-AST. (**d**) ME-AST. (**e**) HCl-AST.

**Figure 3 molecules-26-03569-f003:**
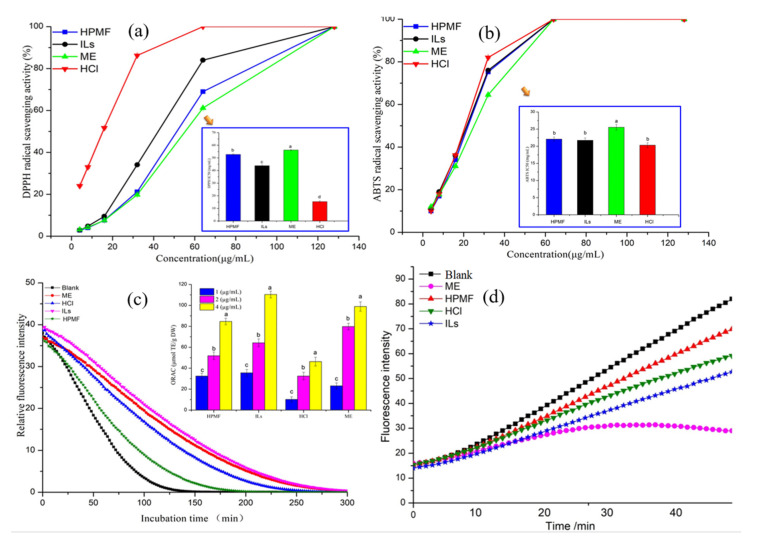
(**a**) Scavenging activity of AST from different extraction methods on DPPH. The inserted figure represents the IC_50_ of DPPH by different extraction methods); (**b**) scavenging activity of AST from different extraction methods on ABTS. The inserted figure represents the IC_50_ of ABTS by different extraction methods); (**c**) effects of AST from different extraction methods on oxygen radical absorbance capacity. The curves are kinetic curves of fluorescence decay induced by AAPH in the presence of AST from different extraction methods, and the inserted figure represents the evaluation of antioxidant activity of four AST at different concentrations. (**d**) Kinetic curves of fluorescence decay of AST from different extraction methods. ANOVA is used and statistically significant differences (*p* < 0.05) between values are indicated by different letters.

**Figure 4 molecules-26-03569-f004:**
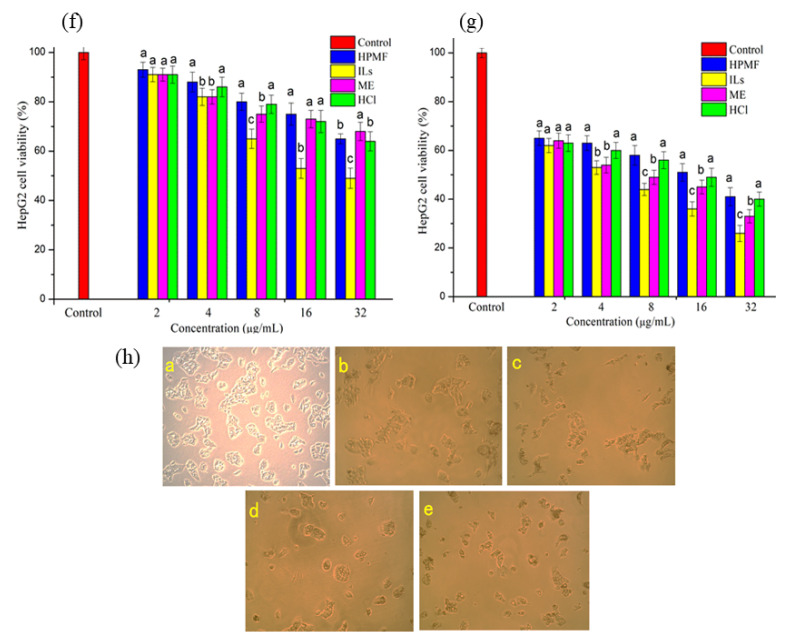
Growth inhibition HepG2 cells of AST by different extraction methods after different incubation times 24 h (**f**) and 48 h (**g**); (**h**) morphologic changes of cells treated with AST after 24 h. (**a**) Control. (**b**) Treated with HPMF. (**c**) Treated with ILs. (**d**) Treated with ME. (**e**) Treated with HCl. ANOVA is used, and statistically significant differences (*p* < 0.05) between values are indicated by different letters.

**Figure 5 molecules-26-03569-f005:**
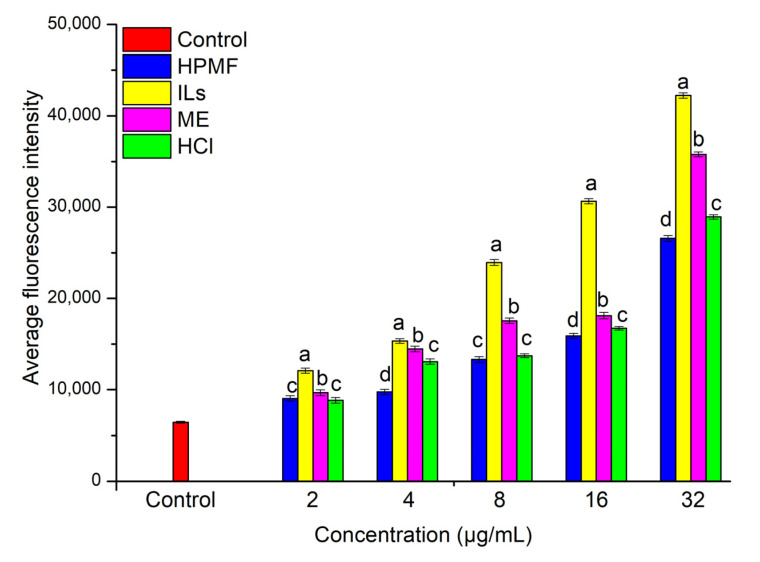
Effects of AST from different extraction methods on accumulation of ROS. ANOVA is used, and statistically significant differences (*p* < 0.05) between values are indicated by different letters.

**Figure 6 molecules-26-03569-f006:**
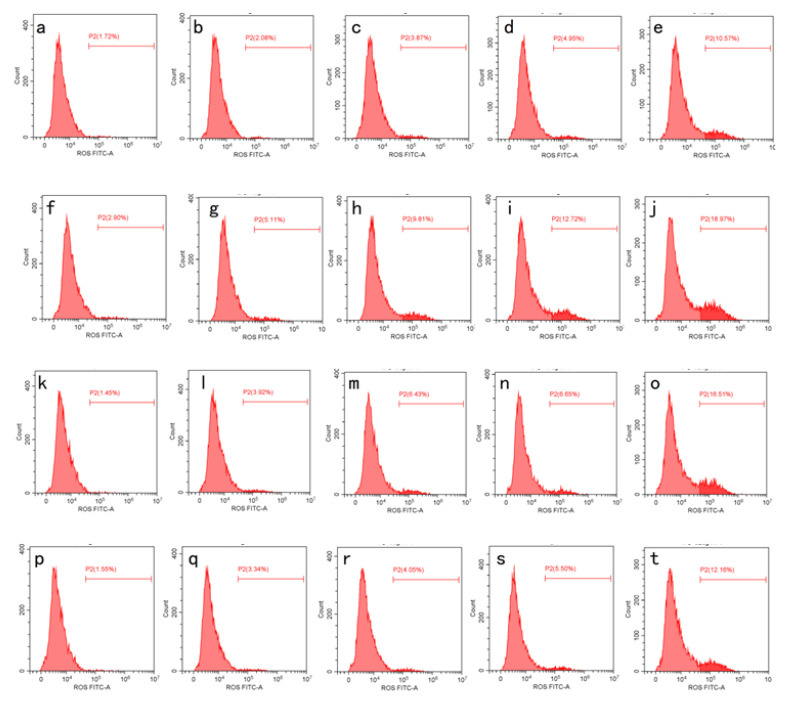
Impact of AST extracted by different methods on generation of ROS. (**a**–**e**) HPMF-AST with the concentrations of 2, 4, 8, 16 and 32, respectively. (**f**–**g**) ILs-AST with the concentrations of 2, 4, 8, 16 and 32, respectively. (**k**–**o**) ME-AST with the concentrations of 2, 4, 8, 16 and 32, respectively. (**p**–**t**) HCl-AST with the concentrations of 2, 4, 8, 16 and 32, respectively.

**Table 1 molecules-26-03569-t001:** Effects of AST from different extraction methods on peroxyl radical scavenging capacity (PSC). ANOVA is used, and statistically significant differences (*p* < 0.05) between values are indicated by different letters.

Extraction Methods	EC_50_ (µM)	PSC (µmoL VCE/µmoL Fruit)
HPMF	11.84 ± 0.54a	10.68 ± 0.82d
ILs	5.52 ± 0.28c	23.55 ± 1.15b
ME	4.28 ± 0.22d	30.37 ± 1.49a
HCl	10.17 ± 0.47b	12.78 ± 0.57c

## Data Availability

Available on request from the corresponding authors.

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
