# Peer review of "Comparison of Different Methods for Extracting the Astaxanthin from Haematococcus pluvialis: Chemical Composition and Biological Activity"

_molecules, 2021, doi:10.3390/molecules26123569_

Round 1

Reviewer 1 Report

The article is interesting, it brings new information on the influence of different techniques of disrupting Haematococcus pluvialis cells on the content of active substances and the activity of extracts.
It is a lengthy study, and the manuscript is well written. Here are some suggestions for further changes.

  1. In Table 1 and Figures 3, 4 and 5, the references to statistics should be completed. The authors are asked to specify the differences indicated by lowercase letters a, b, c ...
    The photos in Figures 1 and 4 can be improved. The photograph of the identification of compounds by TLC (Figure 1) is quite unreadable.
  2. In Table 1 and Figures 3, 4 and 5, the references to statistics should be completed. The authors are asked to specify the differences indicated by lowercase letters a, b, c ...
  3. The authors are kindly asked to refresh the literature. Unfortunately, the cited reports are not the latest, many are from 10 years ago.

Author Response

Reviewers 1

The article is interesting, it brings new information on the influence of different techniques of disrupting Haematococcus pluvialis cells on the content of active substances and the activity of extracts.
It is a lengthy study, and the manuscript is well written. Here are some suggestions for further changes.

  1. In Table 1 and Figures 3, 4 and 5, the references to statistics should be completed. The authors are asked to specify the differences indicated by lowercase letters a, b, c ...
    The photos in Figures 1 and 4 can be improved. The photograph of the identification of compounds by TLC (Figure 1) is quite unreadable.

ResponseThanks for your suggestion, it is corrected

  1. In Table 1 and Figures 3, 4 and 5, the references to statistics should be completed. The authors are asked to specify the differences indicated by lowercase letters a, b, c ...

ResponseThanks for your suggestion, it is corrected

  1. The authors are kindly asked to refresh the literature. Unfortunately, the cited reports are not the latest, many are from 10 years ago.

Response References has been updated.

Reviewer 2 Report

In the presented manuscript, the authors employed different cell disruption techniques (high-pressure micro fluidization, ionic liquids, multi-enzyme, and hydrochloric acid) to extract astaxanthin from H. pluvialis. Further, they examined the impact of different extraction methods on astaxanthin's chemical composition and biological activity. The work is scientifically sound and significant, but I have some objections.

The language and the manuscript style are not so clear, so I recommend some parts to be rewritten, especially the abstract and introduction.  Also, the discussion needs to be improved, with a greater review of the literature.

Determination of Antioxidant Properties of H. pluvialis should be explained in more detail for the sake of completeness.

There are multiple abbreviations introductions, such as AST (line 31) or TLC (line 75); this should be corrected.  

Overall, the manuscript needs an improved presentation of the results and a more conclusive discussion with a better understanding of the purpose of conducted experiments.

Author Response

Reviewers 2

In the presented manuscript, the authors employed different cell disruption techniques (high-pressure micro fluidization, ionic liquids, multi-enzyme, and hydrochloric acid) to extract astaxanthin from H. pluvialis. Further, they examined the impact of different extraction methods on astaxanthin's chemical composition and biological activity. The work is scientifically sound and significant, but I have some objections.

The language and the manuscript style are not so clear, so I recommend some parts to be rewritten, especially the abstract and introduction.  Also, the discussion needs to be improved, with a greater review of the literature.

ResponseThanks. The language has been revised.

Determination of Antioxidant Properties of H. pluvialis should be explained in more detail for the sake of completeness.

ResponseMost of the researchers quoted the methods which are already explained by the previous research articles and e have properly cited these authors.

There are multiple abbreviations introductions, such as AST (line 31) or TLC (line 75); this should be corrected.

ResponseAll the abbreviation has been rechecked (Ctrl +F key) and modified.

Overall, the manuscript needs an improved presentation of the results and a more conclusive discussion with a better understanding of the purpose of conducted experiments.

ResponseResults are already presented in a conclusive way, which will be easy to understand for the readers. Please mention if you need further discussion in which parameters.

Round 2

Reviewer 2 Report

I find the authors did not take the comments seriously and recommend carefully check round 1 of this review. 

Author Response

In the presented manuscript, the authors employed different cell disruption techniques (high-pressure micro fluidization, ionic liquids, multi-enzyme, and hydrochloric acid) to extract astaxanthin from H. pluvialis. Further, they examined the impact of different extraction methods on astaxanthin's chemical composition and biological activity. The work is scientifically sound and significant, but I have some objections.

The language and the manuscript style are not so clear, so I recommend some parts to be rewritten, especially the abstract and introduction.  Also, the discussion needs to be improved, with a greater review of the literature.

ResponseThanks. We have revised the MS completely.

Determination of Antioxidant Properties of H. pluvialis should be explained in more detail for the sake of completeness.

ResponseThe method for determination antioxidant were added, and marked with red color.

There are multiple abbreviations introductions, such as AST (line 31) or TLC (line 75); this should be corrected.

ResponseAll the abbreviation has been rechecked and modified.

Overall, the manuscript needs an improved presentation of the results and a more conclusive discussion with a better understanding of the purpose of conducted experiments.

ResponseThanks. We have revised the MS completely.

Round 3

Reviewer 2 Report

The authors responded to all my requests. I recommend the manuscript for publication in the present form.